# The Quality of Life of Patients with Surgically Treated Colorectal Cancer: A Narrative Review

**DOI:** 10.3390/jcm11206211

**Published:** 2022-10-21

**Authors:** Filip Świątkowski, Tomasz Górnicki, Kacper Bułdyś, Mariusz Chabowski

**Affiliations:** 1Department of Surgery, 4th Military Teaching Hospital, 50-981 Wroclaw, Poland; 2Division of Anaesthesiologic and Surgical Nursing, Department of Nursing and Obstetrics, Faculty of Health Science, Wroclaw Medical University, 51-618 Wroclaw, Poland; 3Student Research Club No 180, Faculty of Medicine, Wroclaw Medical University, 50-367 Wroclaw, Poland; 4Faculty of Medical Sciences and Health Sciences, Kazimierz Pulaski University of Technology and Humanities in Radom, 26-600 Radom, Poland

**Keywords:** quality of life, colorectal cancer, stoma, sex life

## Abstract

Introduction. Quality of life is a topic increasingly being addressed by researchers. Due to the increasing incidence of colorectal cancer, this issue is particularly relevant. Despite the increasing number of publications on this topic each year, it still requires further research. The aim of this study was to analyze the available literature from the past 10 years, addressing the topic of QoL in patients with colorectal cancer which has been treated surgically. Material and methods. This review is based on 93 articles published between 2012 and 2022. It analyzes the impact of socioeconomic factors, the location and stage of the tumor, stoma and the method of surgical treatment on patients’ QoL and sexual functioning. Results. CRC has a negative impact on patients’ financial status, social functioning, pain and physical functioning. Patients with stage II or III cancer have an overall lower QoL than patients with stage I. The more proximally the lesion is located to the sphincters, the greater the negative impact on the QoL. There was a significant difference in favor of laparoscopic surgery compared with open surgery. In patients with a stoma, the QoL is lower compared with patients with preserved gastrointestinal tract continuity. The more time has passed since surgery, the more the presence of a stoma has a negative impact on QoL. Surgery for CRC negatively affects patients’ sex lives, especially in younger people and among men. Conclusions. This study may contribute to the identification of the factors that affect the QoL of patients with surgically treated colorectal cancer. This will allow even more effective and complete treatment, facilitating patients’ return to normal physical, mental and social functioning.

## 1. Introduction

Health-related quality of life (HRQoL) and the analysis of factors influencing it is becoming a topic more and more frequently studied in scientific research. In 2015, 8679 studies on the Quality of Life (QoL) of patients with colorectal cancer (CRC), were published and were available on the PubMed database. By 2020, this number had increased to 13,792 items. Due to their holistic approach to the patient, apart from measurable clinical parameters (e.g., deviations in laboratory test results), attention was also paid to the patient’s mental state and social functioning. Undoubtedly, the disease process affects all spheres of human life—physical, mental and social [1]. 

Due to the high incidence of CRC (second place among women and third place among men), the study of the impact of this cancer on the QoL is an important research problem [2,3]. 

Most studies relate to patients being treated with chemotherapy. There are far fewer publications that analyze this issue in relation to patients treated surgically. 

An answer to the questions of how colorectal cancer affects the QoL of cancer patients, which domains are most affected and what the determinants of QoL in this group of patients are will allow for even more effective and complete treatment. In some cases, perhaps, it can also help reduce the length of hospitalization, thereby reducing the cost of therapy. 

## 2. Material and Methods

The literature search was carried out in March 2022. The electronic databases PubMed and Scopus were used to identify relevant articles. The following combination of search terms was applied to find articles about Quality of Life in Colorectal cancer in PubMed: ((quality of life [Title]) AND ((“Colorectal Neoplasms” [Mesh]) OR (“Colorectal Cancer” [Title]) OR (“Colon Cancer” [Title]))) AND (“surgical procedures, operative” [MeSH Terms] OR “Surgical treatment” [Title]). The following combination of search terms was applied to find articles about Quality of Life in Colorectal cancer in Scopus: TITLE-ABS-KEY (“quality of life” AND “colorectal cancer” AND “surgical treatment”). 

To be included in the review, studies had to assess QoL in CRC patients who had been treated surgically. Furthermore, the article (or at least the abstract) had to be published in English, between 2012 and 2022. Studies with a main focus on methodological aspects, like the validation of an instrument, as well as commentaries, editorials, poster abstracts, case reports and qualitative studies, were not considered. 

The research identified 610 articles, 341 of which were published within the last 10 years. After examining the abstract and full text, 91 articles remained. A schematic process for selecting publications for the review is shown in Figure 1.

### 2.1. Quality of Life

The term “quality of life” was introduced in the medical literature for the first time in the 1960s. Since then, there have been multiple attempts to properly describe this concept [4]. The current WHO definition describes quality of life as an: “individual’s perception of their position in life in the context of the culture and value systems in which they live and in relation to their goals, expectations, standards and concerns” [5]. Health-related quality of life is the part of the quality of life spectrum specifically focused on aspects related to health [6]. 

There are multiple questionnaires that have been developed to assess HRQoL. For example, the Functional Assessment of Cancer Therapy (FACT) and the EORTC Quality of Life Questionnaire (Core) EORTC QLQ-C30. Both of these have specific versions addressed to patients, namely the Functional Assessment of Cancer Therapy—Colorectal (FACT-C) and the EORTC Quality of Life Questionnaire (Colorectal) (EORTC QLQ-C29). Low Anterior Resection Score (LARS) is a specific tool to assess HRQoL in patients with rectal cancer after Low Anterior Resection (LAR) [5].

Selected questionnaires for QoL assessment of patients with CRC are shown in Table 1.

### 2.2. CRC and Its Influence on QoL

According to the American Cancer Society, in 2022 alone, 106,860 new cases of CRC will be diagnosed in America [2]. In 2020, CRC was the third most common type of cancer worldwide [3], with a mean 5-year relative survival rate estimating at around 67% [45]. Over the years, the development of therapies used in treatment has dramatically increased the survival rate of patients from 37.9% in 1998–2002 up to 78.6% in cases of localized disease [46]. The increased number of cancer survivors leads to increased interest in patients’ wellbeing after treatment, with the leading question being how different types of treatment influence patients QoL.

Depending on the stage at which the cancer was diagnosed, CRC significantly influences the QoL of the patient. Analysis conducted in 2021 provides evidence of significantly lower QoL in patients with diagnosed CRC in stage II and III than in stage I, suggesting that the early stage of disease did not change their basic biological functions and activities [9]. There is also a study suggesting the use of questionnaires measuring QoL to predict the survival rate of patients after surgery for CRC [7]. 

The impact of CRC on QoL may also vary depending on the sex of the patient. Women are more likely to have poorer QoL, nonetheless preserving higher sexual functioning scores and higher scores regarding taste than men [26]. Research conducted on the Iranian and Chinese population of patients with CRC showed that CRC impacted the most negatively on the ‘financial status, social function, pain and physical function sections of patients’ EORTC QLQ-C30 [47,48]. Patients with CRC also had lower scores in the sections on social functioning and financial difficulties, but had better physical functioning than the healthy control group [49]. Some analysis provides evidence of CRC having a greater impact on younger patients than older ones [50]. In contrast, there some studies provide data indicating that before beginning the treatment patients reported physical and cognitive functioning comparable to that of the general population, whereas global health, social-, role- and emotional functioning were significantly lower. Also fatigue and insomnia were more common in CRC patients than in healthy people [51]. Data shows that sexual functioning is significantly impaired in CRC patients in comparison to healthy people [52]. Studies provide evidence that personal characteristics and symptoms of distress are factors that impact the QoL [53].

The placement of the tumor can also impact a patient’s QoL. There is evidence of patients reporting worse HRQoL with colon cancer than with rectal cancer. Patients with colon cancer in comparison with patients with rectal cancer had a higher score in the emotional functioning section of the EORTC QLQ-C30 and reported fewer financial difficulties [26]. The placement of the tumor in the rectum may also influence QoL, with lower rectal cancer tending to lead to lower QoL [54]. When it comes to lower rectal cancer, scientists also provided evidence of distal lesions being the reason for lower QoL than in the case of proximal ones [50]. 

### 2.3. Sociodemographic Factors and QoL in CRC Patients 

Two of the main factors taken into account in the literature are age and sex. As mentioned previously, there is a grave concern among these patients about their financial status. Age is an important factor when considering financial status and the type of treatment that will be applied. The type of cancer can also be impactful. Reports suggest that rectal cancer is more problematic in financial terms than colon cancer, while the localization of cancer also has a significant impact [4,26,50]. 

It has been shown that lower socioeconomic groups are likely to have greater financial problems [26]. However, it has been proven that there were no differences among elderly patients’ QoL, whether they had an open or a laparoscopic colectomy. However, there were differences when comparing the QoL of younger patients who had the laparoscopic operation with older patients who underwent the same treatment [55]. These differences were in favor of the younger patients. Older patients with an ostomy may have HRQoL comparable to the normative population or older patients without an ostomy [52]. It is worth mentioning here that in a group of functionally dependent patients, there was a report of clinically relevant improvement of QoL after the patient underwent surgery [11]. 

The aspects of life which are most affected by the occurrence of cancer are: social functioning, financial status and physical activity [47,56]. Interestingly, there is the effect of “rejoice” that is used by researchers to explain an equal or higher QoL result in CRC patients compared to the healthy population [49]. There is a report explaining this effect, and it is worth remembering when assessing QoL [57]. 

The type of surgical procedure influences the QoL of CRC patients [58,59]. Surprisingly it is still unclear whether socioeconomic status affects the QoL of CRC patients with a stoma who survived or did not [60]. A group of frail patients has been shown to improve in emotional functioning at 3 months after the surgery [25]. There are some differences depending on sex. The female gender was associated with a reduction in QoL; men were more susceptible to having worse bowel function than women [61,62].

Evaluating the impact of comorbidities on the quality of life of patients with colorectal cancer treated surgically may be an important line of research. This is of particular importance in an era of increasingly prevalent lifestyle diseases, such as diabetes and obesity. 

### 2.4. The Relationship between QoL and the Chosen Method of Surgical Treatment

Currently, there are several operation types that can be used in CRC treatment. It has been proven that there are no clinically significant differences between laparoscopic and open surgery [63], although the laparoscopic method has a comparable or better survival rate [29]. Interestingly one study stated that after 12 months, global QoL was restored following both methods [64]. 

Considering the QoL of patients, laparoscopic surgery (LS) is better for the patient, as a shorter recovery time is needed, and it causes less severe symptoms [16,22,65,66,67,68,69,70]. Open surgery is more likely to require a stoma, which has a direct negative impact on patients’ QoL [12]. Single-incision laparoscopic surgery (SILS) leads to a better global health score than multiport surgery [39]. A comparison of laparoscopic total mesorectal excision (LaTME) and transanal total mesorectal excision (TaTME) shows that both methods have similar oncological outcomes [71] and do not display any significant differences [43]. TaTME was proven to have acceptable QoL at both 6 and 12 months after the operation [72,73]. 

Endoscopic surgery caused patients to be more afraid of the cancer recurring [24]. Transanal minimal invasive surgery (TAMIS) does not impact the QoL of patients in over a 3-year follow-up period. Moreover, its functional outcome is within acceptable limits and can compete with transanal endoscopic microsurgery (TEM) [74,75,76]. TEM is considered a safe, effective and minimally invasive surgery [77]. There are a few aspects worth noting: sphincter preserving and definitive colostomy made no significant difference to global QoL scores [71]. However, patients with a J-pouch may benefit in the short term with regard to QoL [78].

In the era of rapidly developing robotic surgery, it seems that more research should be done on the impact of this type of surgical treatment on patients’ quality of life.

Enhanced recovery after surgery (ERAS) shows better outcomes than standard interventions [79,80]. A report shows that robot-assisted surgery resulted in improved urogenital function than after laparoscopy [81].

### 2.5. The Impact of a Stoma on QoL 

It is well known that a stoma impacts QoL in CRC patients and that its presence is often counted as a negative outcome. Specific questionnaires are used to assess QoL in CRC patients. The most common of these are SF-36, EORTC QLQ-C30 and EORTC QLQ-CR29 (which is a replacement for EORTC QLQ-CR38) [82]. It was stated that when QoL is being compared between groups it is better to use EORTC QLQ-C30 instead of SF-36 [63]. 

Generally speaking, the presence of a stoma is enough to lower a patient’s QoL, most probably in the area of social functioning [31,48,83]. It has been proven that a stoma may lead to the person quitting work, which can cause financial difficulties. Interestingly, among the results of our research, we could not find any studies which proved that there were financial difficulties in a group of patients with either permanent or temporary ostomies [27,84]. 

However, there are differences between reports on how severe an impact a stoma has on QoL. One study reported that, irrespective of age, the influence was minor. In contrast to this, one of the COCHRANE reviews says that there were no apparent differences in QoL in patients with a permanent stoma and non-stoma patients [15,52]. The timing of the patients’ QoL assessment is very important. It has been proven that the questionnaires completed shortly after surgery did not show statistically significant differences, but over time, it tended towards patients with a stoma having a lower social functioning score [26,85]. Several studies have confirmed that stoma formation affects social functioning or causes psychological problems, for example in Mediterranean and Middle Eastern cultures or the Chinese, Iranian and Korean population [32,83]. It is worth mentioning the response-shift phenomenon in the approach to the patient after CRC surgery, which can occur when a patient has a non-permanent stoma [86]. This phenomenon is of particular interest because it is difficult to study objectively [87,88,89]. Moreover, a study says that there was no significant difference in the global QoL scores in patients with sphincter preservation or a permanent colostomy [71]. It was found that the self-efficacy of patients is not very high, unless appropriate nursing interventions are being used [90]. 

Conducting more in-depth studies on the impact of the type of stoma (colostomy, ileostomy) on patients’ quality of life could provide valuable clinical information.

### 2.6. The Impact of CRC on Sex Life 

Consideration of a patient’s sex life is becoming more and more noticeable in relation to the treatments patients are receiving. There is also a growing interest in how those methods affect them. Unfortunately, most of the commonly used treatment methods negatively affect sex life. 

As mentioned in the previous paragraph, it is worth noting that an assessment of patients’ sexual function score is time-dependent [72,91]. The impact of a stoma has on patients’ lives was analyzed. Irrespective of whether a stoma is permanent or temporary, it causes inconvenience in sexual functioning, in the same was as having a bulge or a hernia does. In addition, no studies have shown an improvement in sexual functioning after colon reconstruction [19,20,33,63,92,93]. However, one publication reports that the presence of a stoma seemingly was not a factor that would adversely affect sexual functions [94]. Surprisingly, one study says that patients who were sexually active before the operation have reported no sexual dysfunction after the treatment [95]. Considering the long-term postoperative period, it is worth mentioning that sexual functioning may decline rapidly [12,62]. 

Males are more often negatively affected by the surgery. However, it has been proven that some surgical approaches have less of an impact on sexual functioning [20,28,37,38,44,68,84,96,97,98,99]. Taking female sexual functions into account, there is a report of them decreasing over time. In contrast, there are also reports of an improvement [20,26,84]. Younger patients are seemingly more negatively affected by surgery than elderly people [49,50]. 

Researchers also point out that there is a problem with collecting data on women’s sex life, mainly because of the low response rate to questionnaires. Moreover, there are reports dedicated exclusively to the assessment of women’s sex life after colorectal surgeries [72,100].

The vast majority of studies on the impact of colorectal cancer on sex life involve heterosexual patients. It seems important to assess the quality of life of patients of all sexual orientations.

## 3. Conclusions

The problem of QoL of patients with CRC is increasingly being addressed by the scientific community. Moreover, this problem is particularly important due to the dynamics of the increase in the incidence of CRC [2,3]. 

It was noted that CRC has a particularly negative impact on patients’ financial status, social functioning, pain and physical functioning [47,48,49]. Younger people are more affected by generally understood problems and difficulties related to the diagnosis and treatment of a neoplastic disease as even surgical treatment of nonagenarian patients can be done safely and without postoperative mortality [50,101].

Overall, QoL deteriorates in women more significantly than in men. The opposite occurs in the context of sexual functioning [26]. 

One of the main factors influencing the QoL of patients is the financial problems associated with the disease [47,48]. This phenomenon affects mainly lower-ranked socioeconomic groups [26]. 

Patients diagnosed with stage II or III cancer have an overall lower QoL than patients with stage I cancer [9]. The more proximally the lesion is located to the sphincters, the greater the negative impact on the QoL [26,50,54]. 

There have also been numerous studies on the impact of the surgical treatment method on the QoL of patients with CRC [13,102,103]. There was a significant difference in favor of laparoscopic surgery compared to open surgery [16,22,65,66,67,68]. The advantage of single-incision laparoscopic surgery (SILS) methods over multiport access has also been demonstrated [39]. The studies also confirmed the positive effect of the ERAS protocol on the course and results of treatment [79,80]. 

In patients with an established stoma, the QoL is lower compared to patients with preserved gastrointestinal tract continuity [27,31,48,83,84]. A stoma impairs social functioning to the greatest extent [103]. It has also been shown that the more time has passed since surgery, the more the presence of a stoma has a negative impact on QoL [26,85]. 

Overall, surgery for CRC negatively affects patients’ sex lives [52]. An important factor that significantly reduces the quality of a patient’s sex life is the necessity to create a stoma [19,20,33,63,92,93]. The surgical procedure affects the quality of sex life to varying degrees, depending on the surgical treatment method chosen [28,37,38,44,68,84,96,97,98,99]. 

## 4. Practical Implications

Any disease, especially one as serious as cancer, will have a negative impact on QoL. Understanding the factors that have the greatest impact on the QoL of patients treated surgically for CRC will allow for more effective treatment, as well as improving of the mental state of patients and facilitating their faster return to satisfactory social functioning. Much has already been done in this area, but there are still aspects that require further research. 

## Figures and Tables

**Figure 1 jcm-11-06211-f001:**
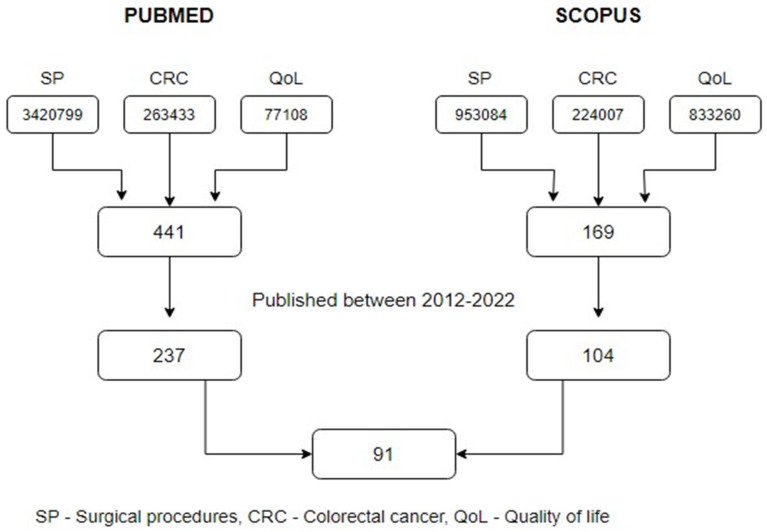
A schematic process for selecting publications for the review.

**Table 1 jcm-11-06211-t001:** Selected questionnaires for quality of life assessments of patients with colorectal cancer.

Lp.	Role	Questionnaire	Abbreviation	Reference
1	Assessment of emotional state	Positive and negative affect schedule	PANAS	[7]
2	Assessment of emotional state	Mood rating scale	MRS	[7]
3	Functionality assessment	Hand Grip Scale	HGS	[8]
4	Functionality assessment	Mini Mental State Examination	MMSE	[9]
5	Functionality assessment	Nottingham Extended Activities of Daily Living Scale	NEADL	[9]
6	Functionality assessment	Functional Assessment of Cancer Therapy—General	FACT-G	[10]
7	Functionality assessment	Barthel Index for Activities of Daily Living	BI	[11]
8	General QOL measurement	European Organization for the Research and Treatment of Cancer Quality of Life Questionnaire—C30	QLQ-C30	[12]
9	General QOL measurement	European Organization for the Research and Treatment of Cancer Quality of Life Questionnaire—CR29	QLQ-CR29	[13]
10	General QOL measurement	European Organization for the Research and Treatment of Cancer Quality of Life Questionnaire—CR38	QLQ-CR38	[14]
11	General QOL measurement	Short Form Health Survey	SF-36	[15]
12	General QOL measurement	EuroQoL-5 Dimensions	EQ-5D	[16]
13	General QOL measurement	Functional Assessment of Cancer Therapy-Colorectal	FACT-C	[9]
14	General QOL measurement	Cleveland Global Quality of Life	CGQL	[17]
15	General QOL measurement	European Prospective Investigation into Cancer and Nutrition	EPIC	[18]
16	General QOL measurement	modified City of Hope Quality of Life-Colorectal	mCOH-QOL-CRC	[19]
17	General QOL measurement	12-Item Short Form Survey	SF-12	[20]
18	General QOL measurement	Karnofsky scale	KPS	[21]
19	General QOL measurement	Edmonton Frail Scale	EFS	[9]
20	General QOL measurement	Échelle de Mesure des Manifestations du Bien-Être Psychologique	EMMBEP	[17]
21	General QOL measurement	Mini Nutritional Assessment	MNA	[9]
22	General QOL measurement	QoL Index	QLI	[22]
23	General QOL measurement	Global QoL Score	GQoLS	[22]
24	General QOL measurement	European Organization for the Research and Treatment of Cancer Quality of Life Questionnaire—PR25	QLQ-PR25	[23]
25	General QOL measurement	European Organization for the Research and Treatment of Cancer Quality of Life Questionnaire—CX24	QLQ-CX24	[23]
26	Psychological aspects	Cancer Worry Scale	CWS	[24]
27	Psychological aspects	Geriatric Depression Scale	GDS	[25]
28	Psychological aspects	Impact of Event Scale—Revised	IES-R	[26]
29	Psychological aspects	Life Orientation Test—Revised	LOT-R	[26]
30	Psychological aspects	Post-traumatic Growth Inventory	PGI	[25]
31	Psychological aspects	Rosenberg’s self-esteem scale	RSS	[27]
32	Psychological aspects	Body-Image Questionnaire	BIQ	[28]
33	Psychological aspects	Questionnaire on distress in cancer survivors	QSC-R10	[29]
34	Psychological aspects	Illness Perception Questionnaire	IPQ-R	[30]
35	Psychological aspects	Acceptance of Illness Scale	AIS	[31]
36	QOL with Ostomy	City of Hope Quality of Life–Ostomy Questionnaire	COH-QOL-Ostomy	[25]
37	QOL with Ostomy	Colostomy Questionnaire	CQ	[32]
38	QOL with Ostomy	Stoma Quality of Life Scale	SQOLS	[33]
39	QOL with Ostomy	Coloplast stoma QoL	CSQoL	[34]
40	Sexual function assessment	International Index of Erectile Function	IIEF-5	[35]
41	Sexual function assessment	International Prostate Symptom Score	IPSS	[36]
42	Sexual function assessment	Female Sexual Function Index	FSFI	[37]
43	Sexual function assessment	McCoy female sexuality questionnaire	MFSQ-9	[38]
44	Social support assessment	Functional Social Support Questionnaire	FFSQ	[19]
45	Social support assessment	Multidimensional Scale of Perceived Social Support	MSPSS	[25]
46	Symptoms assessment	Low Anterior Resection Syndrome	LARS	[27]
47	Symptoms assessment	Wexner/Cleveland Clinic Faecal Incontinence Severity Scoring System	CCIS	[39]
48	Symptoms assessment	Faecal Incontinence Quality of Life Scale	FIQLS	[17]
49	Symptoms assessment	Faecal Incontinence Severity Index	FISI	[26]
50	Symptoms assessment	Gastrointestinal Quality of Life Index	GIQLI	[40]
51	Symptoms assessment	Hospital Anxiety and Depressions Scale	HADS	[41]
52	Symptoms assessment	Vaizey/St. Mark’s score	St. Mark’s	[42]
53	Symptoms assessment	Anorectal Manometry	ARM	[24]
54	Symptoms assessment	Cleveland Clinic Faecal Incontinence Score	CCFIS	[43]
55	Symptoms assessment	Memorial Sloan-Kettering Cancer Center Bowel Function Instrument	MSKCC BFI	[44]
56	Symptoms assessment	Symptoms Distress Scale	SDS	[22]

## Data Availability

The data presented in this study are available on request from the corresponding author.

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
