# Peer review of "The Quality of Life of Patients with Surgically Treated Colorectal Cancer: A Narrative Review"

_jcm, 2022, doi:10.3390/jcm11206211_

Round 1
Reviewer 1 Report
Filip ÅšwiÄ…tkowski et al. made a comprehensive review regarding the quality of life of patients with surgically treated colorectal cancer. The present manuscript is well-written and the English is fluent; and therefore, the readability of the present article is high. However, in-depth perusal of the whole manuscript would reveal it actually conveyed nothing new, because tumor stage, stoma creation and tumor rectal location influencing post-operative quality of life is self-evident.
In regard of the age and MIS approach in the context of quality of life, I would like to recommend 2 additional references to be included in the manuscript before the final acceptance for publication:
1.Asian Journal of Surgery Volume 44, Issue 1, January 2021, Pages 329-333
2.Asian Journal of Surgery Volume 45, Issue 1, January 2022, Pages 208-212
Author Response
In accordance with the comments and suggestions received from reviewers, our manuscript
(jcm-1981271) has been revised.
Detailed responses to Reviewer’s comments are listed below:
Filip ÅšwiÄ…tkowski et al. made a comprehensive review regarding the quality of life of patients with surgically treated colorectal cancer. The present manuscript is well-written and the English is fluent; and therefore, the readability of the present article is high. However, in-depth perusal of the whole manuscript would reveal it actually conveyed nothing new, because tumor stage, stoma creation and tumor rectal location influencing post-operative quality of life is self-evident.
In regard of the age and MIS approach in the context of quality of life, I would like to recommend 2 additional references to be included in the manuscript before the final acceptance for publication:
1.Asian Journal of Surgery Volume 44, Issue 1, January 2021, Pages 329-333
2.Asian Journal of Surgery Volume 45, Issue 1, January 2022, Pages 208-212
Response: Thank you for your evaluation and recommendation. We gladly added to our manuscript suggested publication (reference 42 and 88) to increase value of our work.

Reviewer 2 Report
The Authors should be congratulated for performing this narrative review on the quality of life of patients with surgically treated colorectal cancer. This review selected a total of 91 studies from the literature and evaluated several aspects of quality of life for surgically treated CRC patients (sociodemographic factors, surgical treatment, stoma, and sexual life). The study is very dense of information and could possibly provide a good synthesis to the reader on these aspects which are increasingly becoming the main aspect evaluated in colorectal surgery outcomes. However, because it is a synthesis of many aspects this review cannot provide a specific description and discussion for each item as a systematic review could. In my opinion, the study could be valuable but the authors should change the title to narrative review in order to better describe their study. The conclusion section should be synthesized. Also, it would be great if the authors could add specific pinpointed aspects for future studies for each aspect. The text should also be checked for spelling (i.e. COHRANE -> COCHRANE at line 214) and typos.
Author Response
In accordance with the comments and suggestions received from reviewers, our manuscript
(jcm-1981271) has been revised.
Detailed responses to Reviewer’s comments are listed below:
The Authors should be congratulated for performing this narrative review on the quality of life of patients with surgically treated colorectal cancer. This review selected a total of 91 studies from the literature and evaluated several aspects of quality of life for surgically treated CRC patients (sociodemographic factors, surgical treatment, stoma, and sexual life). The study is very dense of information and could possibly provide a good synthesis to the reader on these aspects which are increasingly becoming the main aspect evaluated in colorectal surgery outcomes. However, because it is a synthesis of many aspects this review cannot provide a specific description and discussion for each item as a systematic review could. In my opinion, the study could be valuable but the authors should change the title to narrative review in order to better describe their study.
Response: Thank you for your recommendation and evaluation. We changed the title in order to better describe the study: “The Quality of Life of Patients with surgically treated Colorectal Cancer: a Narrative Review”
The conclusion section should be synthesized.
Response: The conclusion section has been synthesized according to the suggestion.
Also, it would be great if the authors could add specific pinpointed aspects for future studies for each aspect.
Response: The potentially relevant (in the authors' opinion) possibilities of further research for the aspects analyzed in the article have been added.
„Conducting more in-depth studies on the impact of the type of stoma (colostomy, ileostomy) on patients' quality of life could provide valuable clinical information.”
„In the era of rapidly developing robotic surgery, it seems that more research should be done on the impact of this type of surgical treatment on patients' quality of life.”
„Evaluating the impact of comorbidities on the quality of life of patients with colorectal cancer treated surgically may be an important line of research. This is of particular importance in an era of increasingly prevalent lifestyle diseases, such as diabetes and obesity.”
„The vast majority of studies on the impact of colorectal cancer on sex life involve heterosexual patients. It seems important to assess the quality of life of patients of all sexual orientations.”
The text should also be checked for spelling (i.e. COHRANE -> COCHRANE at line 214) and typos.
Response: The text has been re-checked to eliminate spelling errors and typos. This paper was checked and the language mistakes were corrected by Ms Elaine Horyza, English native speaker and English language teacher at Wroclaw University.
